# Structural and mechanistic insights into the MCM8/9 helicase complex

**Zhuangfeng Weng[1†], Jiefu Zheng[1†], Yiyi Zhou[1], Zuer Lu[2], Yixi Wu[1], Dongyi Xu[2], Huanhuan Li[3]\*, Huanhuan Liang[4]\*, Yingfang Liu[1,3]\***

[1]Shenzhen Key Laboratory for Systems Medicine in Inflammatory Diseases, School of Medicine, Shenzhen Campus of Sun Yat-sen University, Shenzhen, China; [2]State Key Laboratory of Protein and Plant Gene Research, School of Life Sciences, Peking University, Beijing, China; [3]Department of Colorectal Surgery, The Sixth Affiliated Hospital, Sun Yat-sen University, Guangdong Institute of Gastroenterology, Guangdong Provincial Key Laboratory of Colorectal and Pelvic Floor Diseases, Guangzhou, China; [4]Pharmaceutical Sciences (Shenzhen), Sun Yat-sen University, Shenzhen, China

**Abstract** MCM8 and MCM9 form a functional helicase complex (MCM8/9) that plays an essential role in DNA homologous recombination repair for DNA double-strand break. However, the structural characterization of MCM8/9 for DNA binding/unwinding remains unclear. Here, we report structures of the MCM8/9 complex using cryo-electron microscopy single particle analysis. The structures reveal that MCM8/9 is arranged into a heterohexamer through a threefold symmetry axis, creating a central channel that accommodates DNA. Multiple characteristic hairpins from the N-terminal oligosaccharide/oligonucleotide (OB) domains of MCM8/9 protrude into the central channel and serve to unwind the duplex DNA. When activated by HROB, the structure of MCM8/9's N-tier ring converts its symmetry from *C3* to *C1* with a conformational change that expands the MCM8/9's trimer interface. Moreover, our structural dynamic analyses revealed that the flexible C-tier ring exhibited rotary motions relative to the N-tier ring, which is required for the unwinding ability of MCM8/9. In summary, our structural and biochemistry study provides a basis for understanding the DNA unwinding mechanism of MCM8/9 helicase in homologous recombination.

## eLife assessment

This paper presents **important** findings on the hexameric structure of MCM8/9, which potentially explain its role as a DNA helicase in homologous recombination. This **solid** work will be of interest to biologists studying DNA transactions.

## Introduction

DNA helicases play critical roles in multiple cellular processes including DNA replication, transcription, recombination, and repair. Dysfunction of multiple DNA helicases has been correlated with various human diseases including cancers, Bloom Syndrome, Werner Syndrome, Fanconi Anemia, reproductive deficiencies, and infertility, *etc* (*Brosh and Matson, 2020*; *Heyer et al., 2010*). One of the most extensively studied subfamilies of DNA helicases is the minichromosome maintenance (MCM) proteins. In the human genome, the MCM family contains two subgroups of AAA+ ATPase/helicase complexes, consisting of eight MCM members (MCM2-9) (*Maiorano et al., 2006*). Structural studies of these complexes are critical for elucidating their mechanisms.

\*For correspondence:
lihuanhuan665580@126.com (HL);
lianghh26@mail.sysu.edu.cn (HL);
liuyingf5@mail.sysu.edu.cn (YL)

[†]These authors contributed equally to this work

**Competing interest:** The authors declare that no competing interests exist.

The hexameric MCM2-7 complex, also known as a replicative helicase, play a vital role in the initiation and elongation of eukaryotic chromosome replication. Previous structural and biochemical studies have revealed that the MCM2-7 complex exhibits high structural flexibility during DNA replication (*Yuan and Li, 2020*). It can adopt multiple conformations by interacting with its loaders and activators to form different intermediates, including the helicase-loading intermediate Orc-Cdc6-Cdt1-MCM2-7 (OCCM), the inactive MCM2-7 Double Hexamer (DH) and the active replicative helicase complex Cdc45-MCM2-7-GINS (CMG) (*Yuan et al., 2017*; *Miller et al., 2019*; *Li et al., 2015*; *Abid Ali et al., 2016*). For loading onto DNA, the apo MCM2-7 hexamer first binds to the licensing factor Cdt1. In the presence of ATP, the initiators ORC and Cdc6 assemble on the origin DNA to form an active loading platform and load the Cdt1-bound MCM2-7 onto DNA to form the OCCM complex (*Yuan and Li, 2020*). The origin DNA is encircled into the central channel of the MCM2-7 hexamer via the MCM2-MCM5 entry 'gate' (*Samel et al., 2014*). While ATP hydrolysis occurs in OCCM, it seems that Cdt1, Cdc6, and ORC are released in sequential disassembly steps, and then the first loaded MCM2-7 encircling DNA recruits a second ORC-Cdc6 complex, which in turn loads the second Cdt1-bound MCM2-7 to form a head-to-head MCM2-7 DH (*Yuan and Li, 2020*). During double-hexamer assembly, the N-tier ring of MCM2-7 rotates by nearly 30° when aligned with the MCM2-7 hexamer in its loading intermediate OCCM (*Noguchi et al., 2017*). As the MCM2-7 double hexamer on DNA is inactive, lots of regulators along with the helicase activators Cdc45 and GINS complex, are required to convert the double hexamer into two activated CMG replicative helicases that translocate along the leading strands with a bi-directional replication mechanism (*Georgescu et al., 2017*; *Eickhoff et al., 2019*; *Douglas et al., 2018*). The aforementioned structural studies have also revealed that layers of characteristic hairpin loops inside the chamber of MCM2-7 hexamer, including the helix-2 insertion loops (H2I), the presensor-1 (PS1) as well as β-turn motifs of oligosaccharide/oligonucleotide (OB) domains, contribute to DNA binding and separation (*Slaymaker and Chen, 2012*). Additionally, another allosteric communication loop (ACL) conserved in all MCM helicases was also found to be important for helicase activity. The ACL mediates inter-subunit interactions between the N-terminal DNA processing domain and the C-terminal AAA+ motor domain (*Sakakibara et al., 2008*; *Barry et al., 2009*).

MCM8 and MCM9 make up the other helicase complex, which is the homolog of MCM2-7 (*Nishimura et al., 2012*). In contrast to the replicative helicase MCM2-7, MCM8 and MCM9 function as a DNA helicase complex (MCM8/9) in HR-mediated DNA repair for DNA double-strand breaks (DSBs) and DNA interstrand crosslinks (ICLs) (*Lutzmann et al., 2012*). Several studies have suggested that cells lacking *MCM8* and *MCM9* are highly sensitive to DNA cross-linking reagents (*Morii et al., 2019*; *McKinzey et al., 2021*). The *MCM8* and *MCM9* knock-out mice were found to be sterile and revealed gametogenesis deficiency as well as chromosomal damage due to impaired HR (*Lutzmann et al., 2012*). During HR repair, MCM8/9 was rapidly recruited to the DNA damage sites and colocalized with the recombinase Rad51 (*Park et al., 2013*). It also interacted with the nuclease complex MRN (MRE11-RAD50-NBS1), which was required for DNA resection at DSBs to facilitate HR repair (*Lee et al., 2015*). Recently, HROB (also known as C17orf53 or MCM8IP) has been identified as an essential factor in loading the MCM8/9 complex to the sites of DNA damage and stimulating its helicase activity to promote replication fork progression during DNA recombination and replication. Loss of HROB led to HR defects because it is necessary for the recruitment and activation of MCM8/9 (*Hustedt et al., 2019*; *Huang et al., 2020*). The HROB harbors a flexible N-terminal domain (NTD), a central proline-rich region (PRR) and an OB-fold domain at the C-terminus. It was reported that the MCM8/9 binding motif (MBM) is embedded in the PRR region and binds to the NTD domains of MCM8/9 (*Huang et al., 2020*). Although progress has been made in understanding the role of MCM8/9 HR during the past decade, the structure-function relationship, especially the structural characterization of MCM8/9 helicase activity, remains unknown. Notably, MCM8/9 is composed of two subunits, MCM8 and MCM9. It is challenging to extrapolate its role through a simple comparison with MCM2-7, which consists of six subunits.

MCM8 and MCM9 proteins are structurally similar, with an MCM domain at the N-terminus (NTD) and an AAA+ ATPase domain at the C-terminus (CTD). The NTD and CTD are connected by flexible linkers (N-C linkers), which allow the movement of the AAA+ domain during unwinding (*Brewster et al., 2008*). The NTDs of MCM8 and MCM9 share an analogous domain organization and can be divided into three domains: the zinc finger domain (ZF), the helical domain (HD), and the OB domain.

In addition to the AAA+ ATPase domain, MCM8 possesses an additional WHD domain in the CTD while MCM9 contains a putative helix-turn-helix (HTH) domain and an extensional C-tail at the C-terminus (*Griffin and Trakselis, 2019*).

In the previous study, we reported a 6.6 Å cryo-EM structure of the human MCM8/9 (hMCM8/9) NTD ring. We also analyzed the crystal structures of MCM8 and MCM9 NTDs separately, which exhibited conformational changes when assembled into the MCM8/9 hexamer (*Li et al., 2021*). However, the low-resolution structure of the MCM8/9 NTD ring is insufficient to fully illustrate the assembly and activation mechanisms of the MCM8/9 hexamer. Here, we present a near-atomic resolution structure of the chicken MCM8/9 (gMCM8/9) complex and a 3.95 Å cryo-EM structure of the human MCM8/9 NTD ring under HROB induction. Based on these structures, we conducted further investigation on the structural features of the DNA-binding region of MCM8/9 and the conformational changes required for its activation by HROB. Our structural and biochemical studies shed light on the structural and functional relationship of the MCM8/9 helicase.

## Results

### Overall structure of the gMCM8/9 helicase complex

As our previous study shows the hMCM8/9 helicase complex is unstable for high-resolution structure determination, we first solved the cryo-EM structure of the MCM8/9 complex from *Gallus gallus*. (*Gallus gallus*, gMCM8/9; amino acid, gMCM8: 50–830; gMCM9: 1–691) (*Figure 1A*, *Table 1*, Materials and methods). The gMCM8/9 shares up to 80% sequence homology with the hMCM8/9. Based on the electron density map, the structures of the NTD and CTD were reconstructed at the resolutions of 3.66 Å and 5.21 Å, respectively, and then aligned into a composite structure (*Figure 1B, C* and *Figure 1—figure supplement 1*).

The structure reveals that gMCM8 and gMCM9 alternately assemble into a pinwheel-like hexamer with a central channel. The trimerization of the gMCM8/9 heterodimer is in line with our previously reported hMCM8/9 structure at 6.6 Å resolution (*Li et al., 2021*). When we performed structural superposition, the NTD ring of gMCM8/9 superimposes well onto that of hMCM8/9, indicating that eukaryotic MCM8 and MCM9 form a conserved hexamer complex during evolution (*Figure 1—figure supplements 2–4*). The diameter of the NTD ring is ~132 Å and that of the inner channel is ~28 Å which is large enough to accommodate dsDNA to pass through (*Figure 1C* and S5). The assembly pattern of the gMCM8/9 complex with a threefold (*C3*) symmetry axis is unique among MCM helicases. It arranges the three MCM8 and three MCM9 in the same planes, enclosing a channel with multiple symmetrical hairpins. This assembly pattern differs from that of the eukaryotic MCM2-7 heterohexamer with non-symmetry and the archaeal MCM homohexamer with sixfold symmetry (*Li et al., 2015*; *Fletcher et al., 2003*). In addition, a distinguishing feature of the gMCM8/9 is that the helical domain of gMCM8 contains a long α5 helix that protrudes around the hexameric ring (*Figure 1B*).

The domain structures of gMCM8/9's, including the MCM domains, the AAA+ motor domains, and the WH domain, fit well into their electron density maps (*Figure 1—figure supplement 5*). For each gMCM8/9 heterodimer, the positions of characteristic hairpin loops of MCM helicase, such as H2I, PS1, OB hairpins (referred to as OB-hps), and the ACL were well defined. Among them, the OB-hps extend to the central channel, while the ACLs are close to the H2I and PS1 hairpins within the AAA+ domains (*Figure 1B*).

### The assembly of gMCM8 and gMCM9 in the NTD ring

As our structural analysis revealed that the gMCM8/9 hexamer is the trimerization of the MCM8/9 heterodimer, this leads to the formation of three dimer interfaces within each MCM8/9 heterodimer and three putative trimer interfaces between each neighboring MCM8/9 heterodimer (*Figure 2A*). The high-resolution structure of the gMCM8/9 NTD ring allows us to analyze these two types of interfaces in detail, which serve as the structural basis of MCM8/9 assembly.

The dimer interface between NTDs of gMCM8 and gMCM9 is largely composed of the β-strands from the OB domain of gMCM9 and the flanking ZF and OB domain of gMCM8. Both the polar and hydrophobic interactions between respective β-strands are responsible for the binding of the MCM8/9 dimer. For example, the side chains of $R220_{gMCM9}$ and $Q248_{gMCM9}$ form hydrogen bonds with the carbonyl groups in the main chains of $I172_{gMCM8}$ and $T226_{gMCM8}$, respectively. In addition, the main

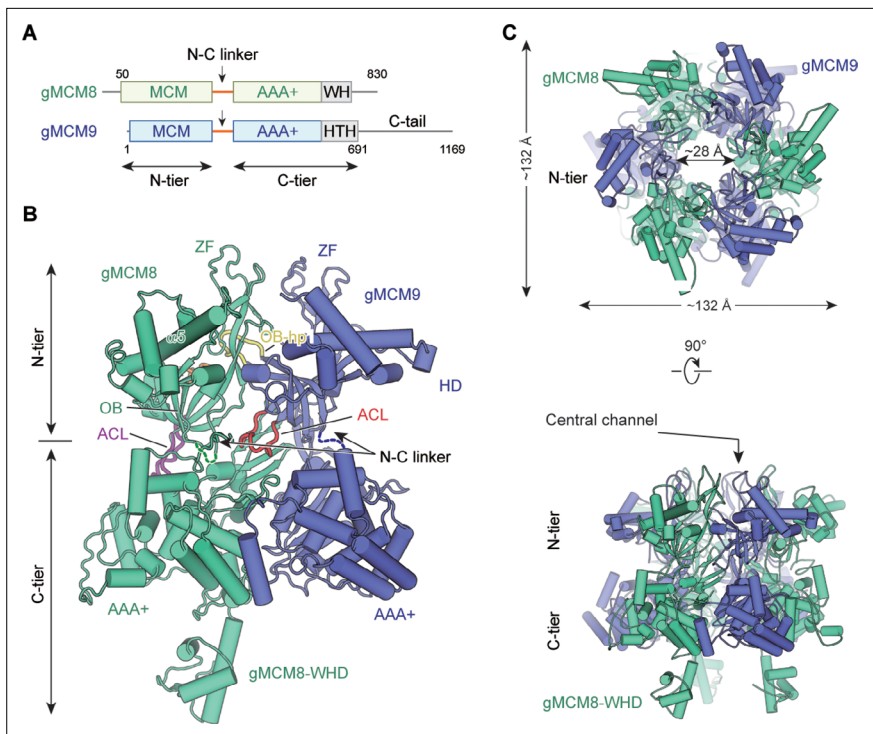

**Figure 1.** Overall structural features of the gMCM8/9 hexamer. (**A**) Domain organization of chicken MCM8 and MCM9. (MCM8, green cyan; MCM9, slate), N-C linker, linkers connecting the N-terminal domain (NTD) and C-terminal domain (CTD) of MCM8 or MCM9; WH, winged helix domain; HTH, helix-turn-helix; the C-tail domain is not included in this study. (**B**) Fold of the gMCM8/9 dimer. The N-C linkers are shown as dotted lines and the OB hairpins OB-hp are highlighted in orange (MCM8) and yellow (MCM9), while the allosteric communication loop (ACL) are highlighted in purple (MCM8) and red (MCM9). The long helix of MCM8 was marked as α5. (**C**) Ribbon diagram showing the top and side views of gMCM8/9 hexamer with threefold symmetry axis.

The online version of this article includes the following figure supplement(s) for figure 1:

**Figure supplement 1.** Resolution evaluation of the gMCM8/9 N-terminal domain (NTD) and C-terminal domain (CTD).

**Figure supplement 2.** Structural superposition of the N-terminal domains (NTDs) of gMCM8/9 and hMCM8/9.

**Figure supplement 3.** Sequence alignments of N-terminal domains (NTDs) of gMCM8 and hMCM8.

**Figure supplement 4.** Sequence alignments of N-terminal domains (NTDs) of gMCM9 and hMCM9.

**Figure supplement 5.** The cryo-electron microscopy (cryo-EM) structure of gMCM8/9.

**Figure supplement 6.** Selected 2D classes of gMCM8/9 showing its typical top and side views.

**Figure supplement 7.** The predicted model uses as a reference for the morphing map of the gMCM8/9 C-terminal domain (CTD).

**Figure supplement 8.** The image processing and 3D reconstruction steps of the gMCM8/9 complex using cryoSPARC.

**Figure supplement 9.** The image processing and 3D reconstruction steps of the gMCM8/9 complex using RELION-3.1.1.

chain of $T119_{gMCM9}$ also forms a hydrogen bond with the neighboring carbonyl group in the side chain of $D228_{gMCM8}$. Furthermore, the benzene ring of $F253_{gMCM9}$ inserts into a hydrophobic pocket surrounded by residues L179, P217, F181, L201, and Y199 from gMCM8 and W250 from gMCM9, which further stabilize the interaction between gMCM8 and gMCM9 (*Figure 2B*, *Figure 2—figure supplement 1A and B*).

Compared to the strong interaction between the dimer interface, the interaction within the trimer interface is weaker. The trimer interface is dominated by electrostatic interaction and hydrogen-bonding interactions. The side chain of $R152_{gMCM8}$ forms salt bridges with both the side chains of $D200_{gMCM9}$ and $D232_{gMCM9}$. In addition, the side chain of $K158_{gMCM8}$ forms a hydrogen bond with the

**Table 1.** Cryo-electron microscopy (Cryo-EM) 3D reconstruction and refinement of the gMCM8/9 complex.

| Data collection and processing | NTD | CTD |
| --- | --- | --- |
| Magnification | 130,000 | |
| Voltage (kV) | 200 | |
| Electron dose (e⁻/Å²) | 50 | |
| Frame | 32 | |
| Under-focus range (µm) | 1.7–2.2 | |
| Pixel size (Å) | 1.0 | |
| Symmetry imposed | *C3* | |
| FSC threshold | 0.143 | |
| **Relion processing** | | |
| Initial particle images (no.) | 965 k | |
| Final particle images (no.) | 144,202 | |
| Global map resolution (Å) | 6.6 | |
| NTD map resolution (Å) | 4.2 | |
| CTD map resolution (Å) | | 5.4 |
| **Cryosparc processing** | | |
| Initial particle images (no.) | 2,134 k | |
| Global map resolution (Å) | 4.31 | |
| Final particle images (no.) | 290,080 | 72,948 |
| NTD map resolution (Å) | 3.66 | |
| CTD mean map resolution (Å) | | 5.21 |
| **Model composition** | | |
| Initial model used (PDB code) | 7DP3 and 7DPD | 3JA8 |
| Non-hydrogen atoms | 13410 | 12,687 |
| Protein | 1698 | 1671 |
| Ligands | 0 | 0 |
| **R.m.s. deviations** | | |
| Bond lengths (Å) | 0.002 | 0.003 |
| Bond angles (°) | 0.593 | 0.843 |
| **Validation** | | |
| MolProbity score | 1.73 | 2.25 |
| Clashscore | 8.36 | 29 |
| Rotamers outliers (%) | 0.00 | 0.00 |
| **Ramachandran plot** | | |
| Favored (%) | 95.64 | 95.61 |
| Allowed (%) | 4.3 | 4.33 |
| Outliers (%) | 0.06 | 0.06 |

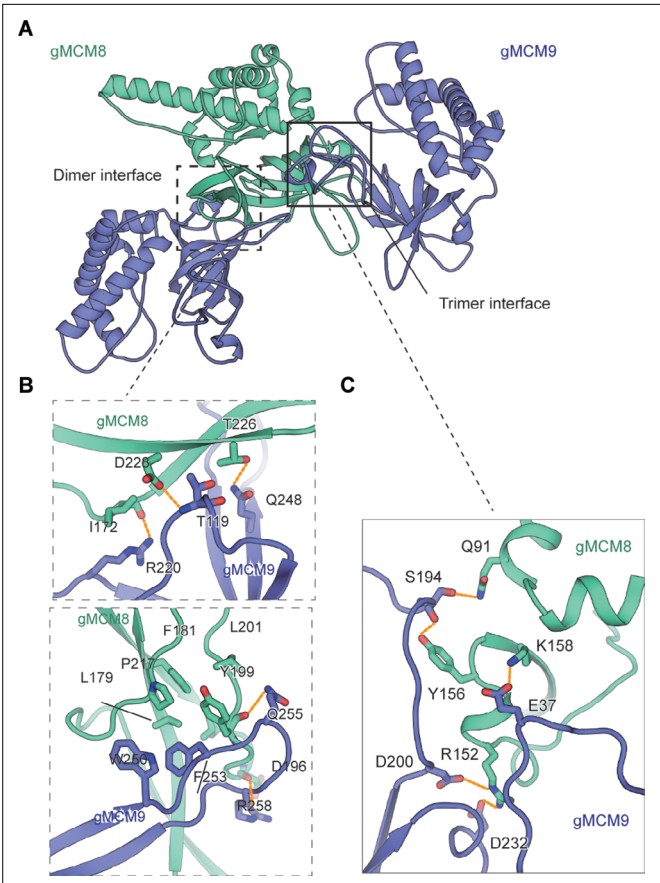

**Figure 2.** The assembly analysis of the gMCM8/9 N-terminal domain (NTD) ring. (**A**) The cutoff structure of the gMCM8/9 NTD presents in the cartoon. The dimer interface and trimer interface were indicated by boxes with dotted lines and solid lines, respectively. (**B, C**) The dimer interface (**B**) and trimer interface (**C**) were mediated by hydrophobic interaction and polar interactions. The interaction details between gMCM8 and gMCM9 in two interfaces are shown in stereo view.

The online version of this article includes the following figure supplement(s) for figure 2:

**Figure supplement 1.** Representative regions of the cryo-electron microscopy (cryo-EM) structure of gMCM8/9 N-terminal domain (NTD).

---

side chain of E37$_{gMCM9}$ (*Figure 2C*, *Figure 2—figure supplement 1C and D*). Based on the preceding analysis, we assume that opening a 'gate' on the trimer interface for DNA loading will be easier than on the dimer interface. Taken together, these interactions facilitate and stabilize the assembly of the NTD hexameric ring of gMCM8/9.

## Structural comparison of MCM8/9 and MCM2-7

To explore the functional state of gMCM8/9, we performed structural comparisons between gMCM8/9 and various MCM2-7-containing complexes. We first compared the gMCM8/9 structure with MCM2-7 in the OCCM complex by aligning their respective C-tier rings. As shown in *Figure 3A*, the NTDs of gMCM8/9 fit well with those of MCM7, MCM3, and MCM5, but rotated slightly in the following MCM2, MCM6 and MCM4. This minor variation is due to the opening 'gate' of the MCM5-MCM2 interface in the OCCM complex. When superimposing the hexameric gMCM8/9 complex onto one hexamer of the MCM2-7 DH, we found that the two C-tier rings aligned well while their N-tier rings had a ~30° clockwise rotation (*Figure 3B and C*). Such rotation could be caused by the two hexameric complex's different assembly patterns. A more plausible explanation is that the assembly of the MCM2-7 DH induces the conformational change in its N-tier ring (*Noguchi et al., 2017*). We also compared the gMCM8/9 structure with that of MCM2-7 in the apo CMG complex and found that the NTD ring has a similar translation and rotation but slighter than in comparison with MCM2-7 DH

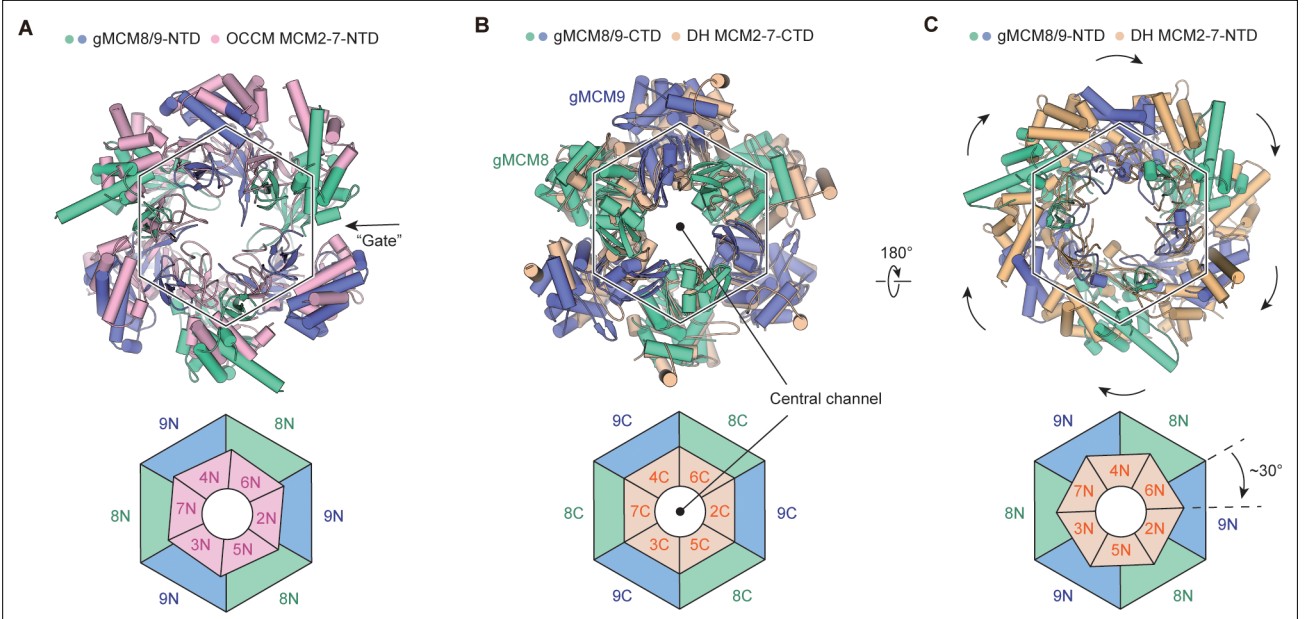

**Figure 3.** Structural comparison of gMCM8/9 with MCM2-7-containing intermediates. (**A**) The gMCM8/9 hexamer (green cyan and slate) was aligned to the MCM2-7 hexamer (light pink) from the Orc-Cdc6-Cdt1-MCM2-7 (OCCM) complex. The N-tier rings comparison are presented here with a slight rotation beginning at MCM5-MCM2 'Gate' and the following MCM6 and MCM4. The 'Gate' was indicated by the black arrow. (**B, C**) Structural superposition of the gMCM8/9 hexamer to the MCM2-7 double hexamer (wheat) by aligning their respective C-tier ring. The bottom view (**B**) and top view (**C**) are shown respectively. Note the ~30° clockwise rotation of the gMCM8/9 N-tier ring compared to that of MCM2-7.

(data not shown). As the NTD ring of MCM2-7 that transits from the DH to the CMG just requires a small domain translation (*Yuan et al., 2016*). Overall, our structure of gMCM8/9 was conformationally similar to the structure of the MCM2-7 complex in its loading intermediate OCCM rather than that in the DH. This is also consistent with previous findings that the recombinant MCM8/9 complex alone exhibited limited helicase activity in vitro (*Huang et al., 2020*). Therefore, we presume the current structure of gMCM8/9 may represent a loading state of MCM8/9 helicase.

## HROB activates MCM8/9 probably by inducing a conformational change of its NTD ring

To explore the activation mechanism of hMCM8/9, we examined its activity in the presence of OB-fold containing protein HROB (*Figure 4A*). Using a helicase assay, we found that the HROB-MBM motif was unable to stimulate the helicase activity of hMCM8/9. However, its CTD, which includes the OB-fold domain, could significantly activate hMCM8/9 (*Figure 4B*). The HROB-CTD not only directly interacts with hMCM8/9 NTD, but also forms a complex with it, as confirmed by the GST pulldown assay (*Figure 4C*).

We collected the complex formed by the HROB-CTD and hMCM8/9 NTD proteins described above and reconstructed their structure using cryo-electron microscopy (*Figure 4D*, *Table 2*, Materials and methods). Unfortunately, the density map of HROB-CTD was not observed in the reconstruction. And the 3.95 Å structure of the hMCM8/9 NTD ring was reconstructed with *C1* symmetry, in contrast to the structure of gMCM8/9, which has *C3* symmetry (*Figure 4—figure supplement 1*). We also found that the structure of the hMCM8/9 NTD ring underwent a large conformational change when compared to the previously described hMCM8/9 NTD (*Li et al., 2021*). As a result, the two conformations were designated as Conformation I (hMCM8/9 NTD at 6.6 Å) and Conformation II (hMCM8/9 NTD at 3.95 Å) (*Figure 4E and F*). In line with our observation, when superimposed these two conformations of hMCM8/9 NTD, we found that one of the trimer interfaces in Conformation II was expanded by ~5 Å (*Figure 4G*). This expansion further pushed its neighboring MCM8/9 heterodimers to shift to either side. The left dimer translates downward by ~8 Å and tilts by ~6° while the right one shifts by ~7 Å and tilts by ~8° (*Figure 4I and J*). Interestingly, the ZF of MCM8 shifts by ~3 Å to the left, and the ZF of MCM9 shifts by ~4.5 Å to the right, resulting in an overall expansion of ~6 Å (*Figure 4G*).

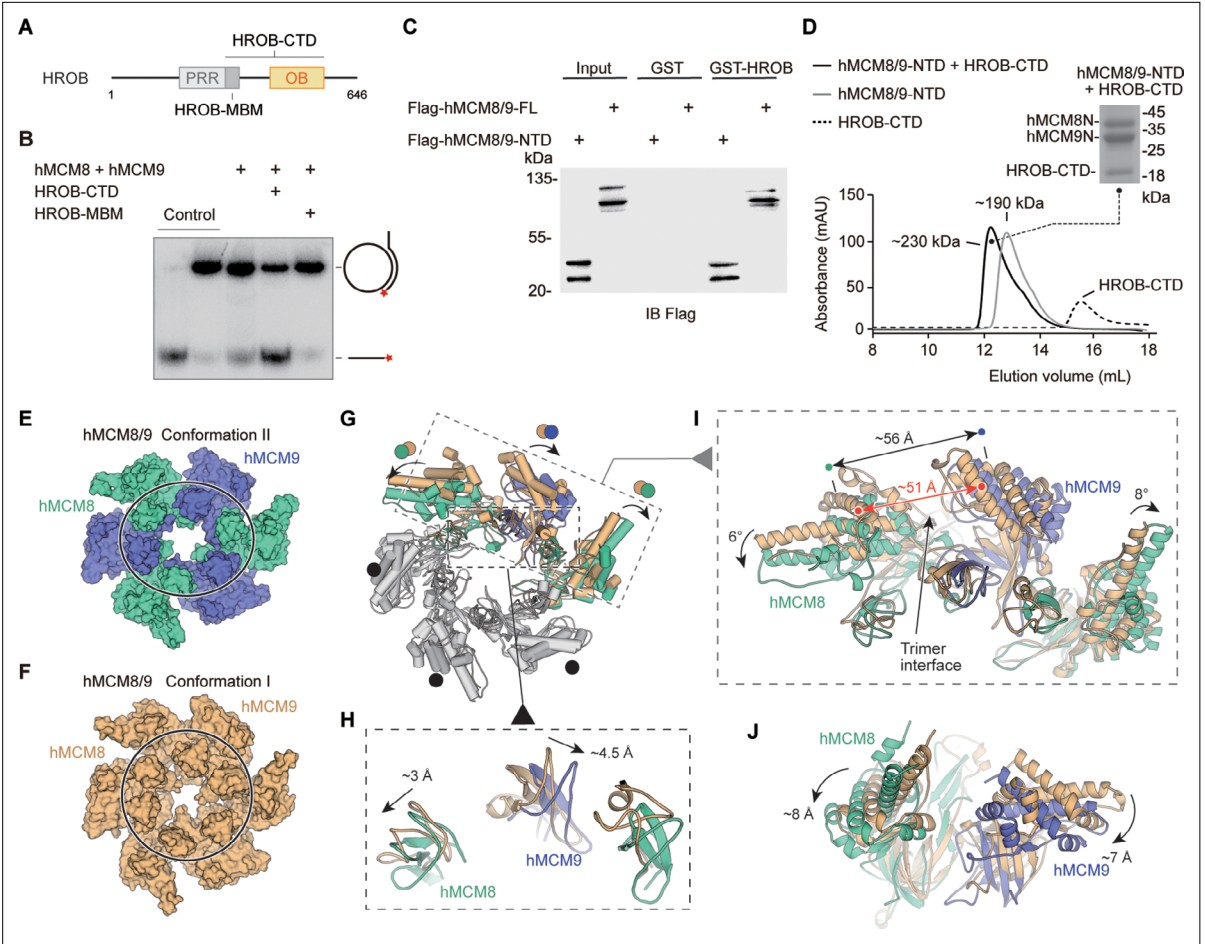

**Figure 4.** HROB induces conformational change of hMCM8/9 complex. (**A**) Domain organization of human HROB. PRR, proline-rich region, gray; MBM, MCM8/9 binding motif, dark gray; OB, OB-fold domain, yellow. The HROB-CTD consisting of the HROB-MBM and oligosaccharide/oligonucleotide domain (OB) is indicated. (**B**) The OB domain is required for MCM8/9 helicase in DNA unwinding. Representative autoradiograph of the DNA unwinding reaction was conducted using a $^{32}$P-labeled ssDNA oligo annealed to the M13mp18 as DNA substrate in the presence of HROB-MBM or HROB-CTD with purified hMCM8/9. Control, DNA substrate without proteins. (**C**) HROB-CTD interacts with MCM8/9-NTD. Detection by western blot of MCM8/9-FL or MCM8/9-NTD co-precipitated by bead-bound GST or GST-HROB. (**D**) The diagram of gel filtration shows the co-purified protein complex of HROB-CTD and MCM8/9-NTD. Their complex is also indicated in SDS-PAGE. (**E**) Reconstructed cryo-EM map of MCM8/9 NTD ring in Conformation II. MCM8, green cyan; MCM9, slate. The shape of the channel is oval. (**F**) The structure of hMCM8/9-NTD from the previous study. MCM8/9, orange. (**G–J**) Structural superposition of the hMCM8/9-NTD in Conformation I and Conformation II. The translations of the Zinc fingers were shown in (**H**). The ~5 Å expansion of the trimer interface of hMCM8/9-NTD in Conformation II was shown in (**I**). The shift distance and angles of the MCM8/9 heterodimer were also shown in (**I**) and (**J**), respectively.

The online version of this article includes the following source data and figure supplement(s) for figure 4:

**Source data 1.** The oligosaccharide/oligonucleotide (OB) domain is required for MCM8/9 helicase in DNA unwinding.

**Source data 2.** HROB-CTD interacts with MCM8/9-NTD.

**Figure supplement 1.** Resolution evaluation of the hMCM8/9 NTD Conformation II.

**Figure supplement 2.** The image processing and 3D reconstruction steps of the N-terminal domain (NTD) ring of hMCM8/9 Conformation II using cryoSPARC.

These changes are consistent with our analysis for the trimer interface of gMCM8/9, which is easier to open for accommodating DNA.

Notably, the N-tier ring of MCM2-7 from the CMG-DNA complex translates laterally by ~12 Å and tilts by ~12° when compared to the structure of MCM2-7 DH (*Noguchi et al., 2017*). Without DNA, the C-tier ring of MCM2-7 from the apo CMG also adopts two different conformations including tilted and untilted, indicating a conformational change of MCM2-7 during activation (*Yuan et al., 2016*). Based on the above observations and the stimulation effect of HROB on the helicase activity

**Table 2.** cryo-electron microscopy (Cryo-EM) 3D reconstruction and refinement of the hMCM8/9 N-terminal domain (NTD) ring.

| Data collection and processing | |
| --- | --- |
| Magnification | 105,000 |
| Voltage (kV) | 300 |
| Electron dose (e⁻/Å²) | 50 |
| Frame | 32 |
| Under-focus range (µm) | 1.7–2.2 |
| Pixel size (Å) | 0.827 |
| Symmetry imposed | *C1* |
| FSC threshold | 0.143 |
| **Cryosparc processing** | |
| Initial particle images (no.) | 1855 k |
| Final particle images (no.) | 95,257 |
| Global map resolution (Å) | 3.95 |
| **Model composition** | |
| Initial model used (PDB code) | 7DP3 and 7DPD |
| Non-hydrogen atoms | 13,878 |
| Protein and DNA residues | 1749 |
| Ligands | 0 |
| **R.m.s. deviations** | |
| Bond lengths (Å) | 0.003 |
| Bond angels (°) | 0.682 |
| **Validation** | |
| MolProbity score | 1.99 |
| Clashscore | 19.18 |
| Rotamers outliers (%) | 0.06 |
| **Ramachandran plot** | |
| Favored (%) | 96.6 |
| Allowed (%) | 3.28 |
| Outliers (%) | 0.06 |

of MCM8/9, we propose that HROB activates MCM8/9 by inducing a conformational change of its NTD ring.

## Spatial distribution and functional role of the OB-fold hairpins

As the hairpin loops of OB-fold domains are essential for DNA binding and/or unwinding in the reported hexameric MCM helicase complexes (*Slaymaker and Chen, 2012*), we analyzed the two types of hairpin loops from OB domains in the NTD rings of gMCM8/9 and hMCM8/9. The first type of OB hairpins of gMCM8 and gMCM9 (refer to OB-hp) protrude into the central chamber in a threefold symmetry manner and narrow down the channel (*Figure 5A*). In particular, the OB-hp of gMCM8/9 forms two layers. The three OB-hp of gMCM8 form the upper layer and the three gMCM9 OB-hp comprise the lower layer (*Figure 5C*). Importantly, the tip of the gMCM8 OB-hp contains several positively charged residues (K337, K339, K342) while the gMCM9 OB-hp contains continuous polar residues (H254, Q255, D256) (*Figure 5D*), which are conserved in hMCM8/9 (R345, K347, K350

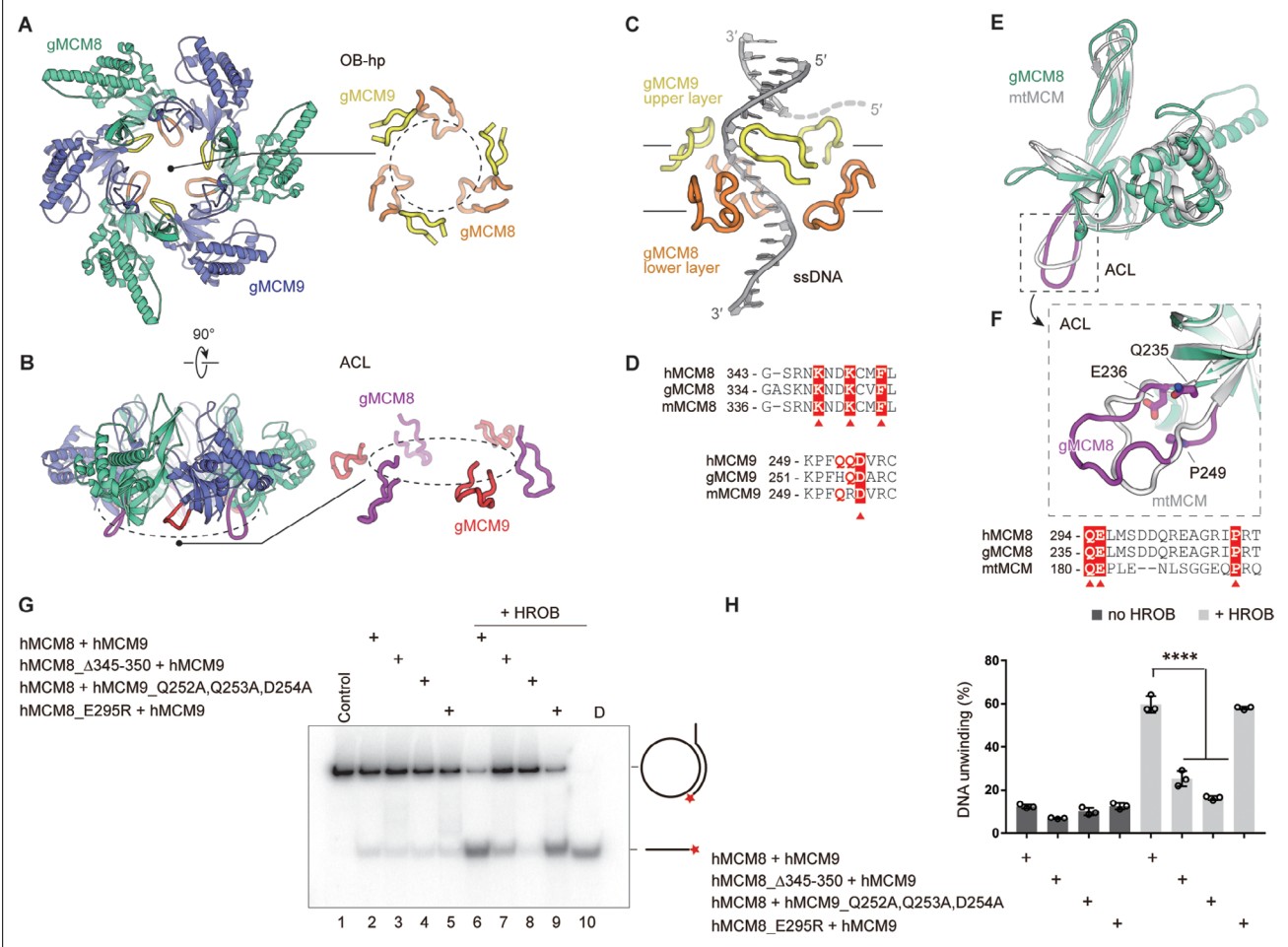

**Figure 5.** The spatial distribution of OB-fold hairpins and their functional roles in DNA unwinding. (**A, B**) Top and side views of the structure of the gMCM8/9 NTD ring. OB hairpins (OB-hps) of gMCM8 and gMCM9 are highlighted in orange and yellow, respectively (**A**); Allosteric communication loops (ACLs) are highlighted in purple and red as indicated (**B**). (**C**) Contacts between OB-hps and forked DNA are illustrated by fitting a forked DNA fragment into the hexameric gMCM8/9 central channel. While encircling the DNA, the OB-hps of the gMCM9 form the upper layer and that of gMCM8 constitute the lower layer. (**D**) Sequence alignments of the OB-hps of MCM8 and MCM9 from different species. h, human; m, mouse; g, chicken (*Gallus gallus*). The highly conserved residues are labeled with red triangles. (**E, F**) Structure superposition of the gMCM8 NTD/mtMCM NTD (gMCM8, green cyan; mtMCM, PDB: 1LTL, gray). The ACL is highlighted by purple (gMCM8) (**E**). Structure-based alignment of minichromosome maintenance (MCM) from different species is shown below the structures and key residues are labeled with red triangles (**F**). (**G**) Representative autoradiograph of the DNA unwinding reaction was conducted using a $^{32}$P-labeled ssDNA oligo annealed to the M13mp18 as DNA substrate in the presence of HROB with purified MCM8/9 or mutants as indicated. (**H**) Graphical representation of the percentage of DNA unwinding in reactions conducted as in (**G**). The mean ± SD of three independent experiments is presented. Statistical analysis was conducted using one-way ANOVA (\*\*p<0.01, \*\*\*p<0.001).

The online version of this article includes the following source data and figure supplement(s) for figure 5:

**Source data 1.** The helicase activities of MCM8/9 OB loop mutants.

**Figure supplement 1.** SEC profiles of wild-type (WT) and OB hairpins (OB-hps) mutants of MCM8/9 complex.

in hMCM8 and Q252, Q253, D254 in hMCM9). As seen in the recently reported cryo-EM structure of the *S. cerevisiae* CMG on a forked DNA, certain conserved residues, such as K364 and K367 from MCM7 OB hairpin, R449 and R451 from MCM4 OB hairpin loop and polar residues Q308 and N307 from MCM3 OB-fold domain, are involved in the interaction with forked DNA and may contribute to its separation (*Yuan and Li, 2020*). We artificially docked a forked DNA into the central channel to generate a gMCM8/9-DNA model and found that the OB-hps of gMCM8 are capable to close contact with it and insert their highly positively charged terminal loops into the major or minor grooves of the DNA strand, implying that they could be involved in substrate DNA processing and/or unwinding (*Figure 5C*).

The OB domains of gMCM8 and gMCM9 also harbor another type of interesting hairpin loop known as the allosteric communication loops (ACLs) (*Figure 5B*). These loops are located on the opposite side of the OB-hp but extend to the CTD ring of gMCM8/9. Several studies have found that the ACLs of *M. thermautotrophicus* MCM (mtMCM) and *Sulfolobus solfataricus* MCM (SsoMCM) helicases facilitate interactions between NTDs and AAA+ catalytic domains of hexameric helicases as well as modulate the positioning of the OB-hp and then regulate the helicase activities (*Sakakibara et al., 2008*; *Barry et al., 2009*). As shown in *Figure 5E and F*, the ACLs show a high degree of conservation when performing the superposition of gMCM8 NTD onto the structure of mtMCM (*Sakakibara et al., 2008*). The sequence alignment of the ACLs also highlights some key residues including Q235, E236 and P249 in gMCM8 (Q294, E295, P308 in hMCM8) that are conserved in mtMCM and important for the mtMCM helicase activity (*Figure 5E and F Sakakibara et al., 2008*). Thus, we propose that the ACLs are required for the helicase activity of MCM8/9.

To investigate the role of OB domain hairpins in regulating the helicase activities of MCM8/9, we deleted the tip residues (345-RNKNDK-350 in hMCM8) to shorten the OB-hp or mutated the polar residues (Q252, Q253, D254 in hMCM9) to change the charge property of OB-hp. Then we performed an in vitro helicase assay based on a plasmid-based substrate (*Lee and Hurwitz, 2001*; *Traver et al., 2015*), using the recombinant hMCM8/9 complex and corresponding mutants (*Figure 5—figure supplement 1*). In the helicase assay, we observed that both the wild-type (WT) hMCM8/9 complex and its mutants only exhibited limited DNA unwinding activity. In the presence of HROB, the WT hMCM8/9 exhibited ~ sixfold stimulation of the helicase activity, while the mutants deleting the tip residues of the hMCM8 OB-hp or mutating that of hMCM9 OB-hp both showed decreased helicase activities. However, the hMCM8 ACL mutant (hMCM8_E295R) corresponding to that in mtMCM had no significant change even in the presence of HROB, implying that the MCM8's NTD doesn't interact with its CTD directly through ACL hairpins (*Figure 5G, H*). Collectively, the above structural analyses and biochemical experiments suggest that the OB-hps of MCM8/9 play an important role in DNA unwinding.

## Structural dynamic study of the gMCM8/9 CTD ring

Since the C-tier of MCM2-7 undergoes conformational change coupling with ATP hydrolysis (*O'Donnell and Li, 2018*), we propose that the CTD AAA+ domain of MCM8/9 is also mobile for DNA unwinding, serving as a motor domain. Notably, the electron density of the gMCM8/9 CTD ring is a little blurry and its structure was reconstructed at 5.21 Å (*Figure 1—figure supplements 1 and 6*). On the basis of our cryo-EM dataset, we performed structural flexibility analyses using multi-body refinement to investigate the dynamics of the gMCM8/9 complex. In this method, we used separate focused refinements and generated movies to describe the most important motions on the relative orientations of the rigid bodies, which allows us to gain insight into the molecular motions of gMCM8/9, particularly regarding the structural flexibility of its CTD ring (*Nakane and Scheres, 2021*; *Liu et al., 2019*; *Singh et al., 2020*). As expected, twelve components were generated, each with a discrete number of independently moving bodies, providing a characterization of the motions in the gMCM8/9 CTD ring. As shown in the *Video 1*, the CTD AAA+ motor ring reciprocally rotates right and left relative to the NTD ring. It would be very interesting to see similar rotary movement along DNA, which may underlie the translocation/unwinding mode for MCM8/9 helicase.

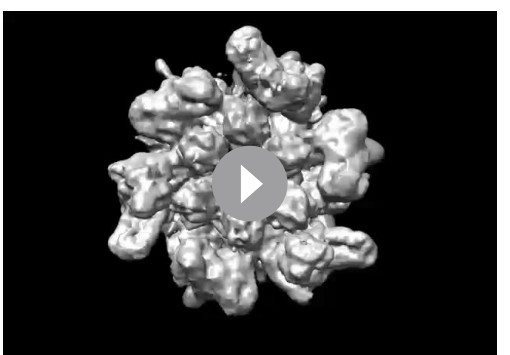

**Video 1.** Structural flexibility analyses of the gMCM8/9 C-terminal domain (CTD) ring.
https://elifesciences.org/articles/87468/figures#video1

## The N-C linkers are required for MCM8/9's helicase activity

Unlike MCM8/9, such rotary motion between NTD and CTD rings has yet to be observed in the structures of MCM2-7. We analyzed the linkers connecting the NTD and CTD (N-C linkers) of MCM8/9 and found that these N-C linkers, which are 30–40 residues long, are longer than those in MCM2-7. In MCM2-7, the N-C linker of MCM3 is the shortest, which may restrict the rotary

motion of MCM2-7 (*Figure 6—figure supplement 1*). To test whether the N-C linkers are crucial for the relative motion between N-tier and C-tier rings of MCM8/9, we shortened the N-C linkers by deleting several amino acids in loop regions of hMCM8 (MCM8Δ369–377) or hMCM9 (MCM9Δ283–287) (*Figure 6—figure supplement 2*). In the helicase assay in vitro, we found that both single and combined deletions have significantly lower unwinding ability than the WT MCM8/9 complex. The presence or absence of HROB had no effect on this result, indicating that the N-C linkers are required for the MCM8/9 helicase activity (*Figure 6A and B*).

We next studied whether the N-C linkers are important for the in vivo functions of MCM8 or MCM9. To do so, we employed a chemoresistance assay because MCM8 and MCM9 had been reported to mediate cellular resistance to the DNA cross-linking reagent cisplatin, which induces DNA interstrand crosslinks that are mostly repaired by HR (*Nishimura et al., 2012*; *Morii et al., 2019*; *Kanemaki, 2013*). As shown in the cell-viability assay, we found a notable reduction in survival of *MCM8*- or *MCM9*- deficient DT40 cells following cisplatin treatment compared to the WT DT40, indicating that the KO cells exhibit significant sensitivity to cisplatin. Expression of WT MCM8 or MCM9 cDNAs in their respective KO cells almost completely complemented chemoresistance. In contrast, the expression of their N-C linker deletion mutants had minimal complementary effects (*Figure 6C, D, E and F*). These results suggest that the N-C linkers of MCM8/9 are required for its cellular resistance to DNA-damaging reagents.

## Discussion

In this study, we reconstructed two structures of the MCM8/9 helicase complex at near-atomic resolution. We found that MCM8 and MCM9 form a single heterohexameric ring with a centric channel that allows the DNA to pass through. The OB hairpin loops within the centric channel may be involved in interacting with and separating substrate DNA. The N-C linkers of MCM8/9 play important roles in helicase activity. Notably, the regulator HROB induces a conformational change of the MCM8/9 NTD ring, which may underlie the helicase activation.

MCM8/9 and MCM2-7 are both MCM proteins with highly conserved domain organization and sequence similarity (*Maiorano et al., 2006*; *Maiorano et al., 2005*). Compared with the heterohexameric structure of MCM2-7, the MCM8/9 hexamer is arranged in an alternate mode with the threefold symmetry axis. Interestingly, it was reported that the *Escherichia coli* helicase DnaB also exists in a quaternary state with *C3* or *C6* symmetry and can change the symmetry states reversibly depending on the pH value of the buffers or the presence of nucleotides (*Donate et al., 2000*). It has been proposed that the symmetry transition plays a role in the loading of regulators on DNA and is directly linked to DNA translocation (*Yang et al., 2002*; *Núñez-Ramírez et al., 2006*). Similarly, we doubt that MCM8/9 changes its symmetry from *C3* to *C1* symmetry in the presence of HROB. In other words, helicases may orchestrate the inner channels by controlling the relative positions of subunits, assisting in DNA loading and unwinding.

Taking inspiration from the structural and unwinding mechanisms of MCM2-7, we also investigated the key elements of MCM8/9 involved in DNA interaction or separation. In the central channel of the MCM2-7 complex, several structural elements involved in DNA binding and strand separation were illustrated by a series of CMG structures complexed with different types of DNA (*Yuan and Li, 2020*). According to recent research, the OB hairpin loops of MCM3, MCM4, MCM6, and MCM7 form 'dam-and-diversion tunnel' to block and divert the lagging strand from the leading strand (*Yuan and Li, 2020*). For the MCM8/9 complex, the OB-hps of MCM8 and MCM9 are essential for DNA unwinding. They form a two-layered structure with threefold symmetry. In which, three OB-hps of MCM9 form the upper layer, while three OB-hps of MCM8 form the lower layer, suggesting a synergistic mechanism for MCM8/9 OB hairpins to interact with DNA. When modeled a forked DNA into the channel, the OB-hp loops make very close contact with the DNA strand that passes through the channel, capable of inserting their loops into the major or minor grooves of the DNA in a sequential manner. While in the Conformation II of hMCM8/9 NTD ring, the OB-hps of hMCM8 and hMCM9 lose their *C3* symmetry without significant conformational change. This symmetry transition could be caused by HROB while inducing trimer interface expansion. In addition, the highly conserved residue F363 from the OB hairpin loop of MCM7 serves as a strand separation pin to unwind forked DNA via making a π-π interaction with DNA (*Baretić et al., 2020*). The highly conserved phenylalanine was also found

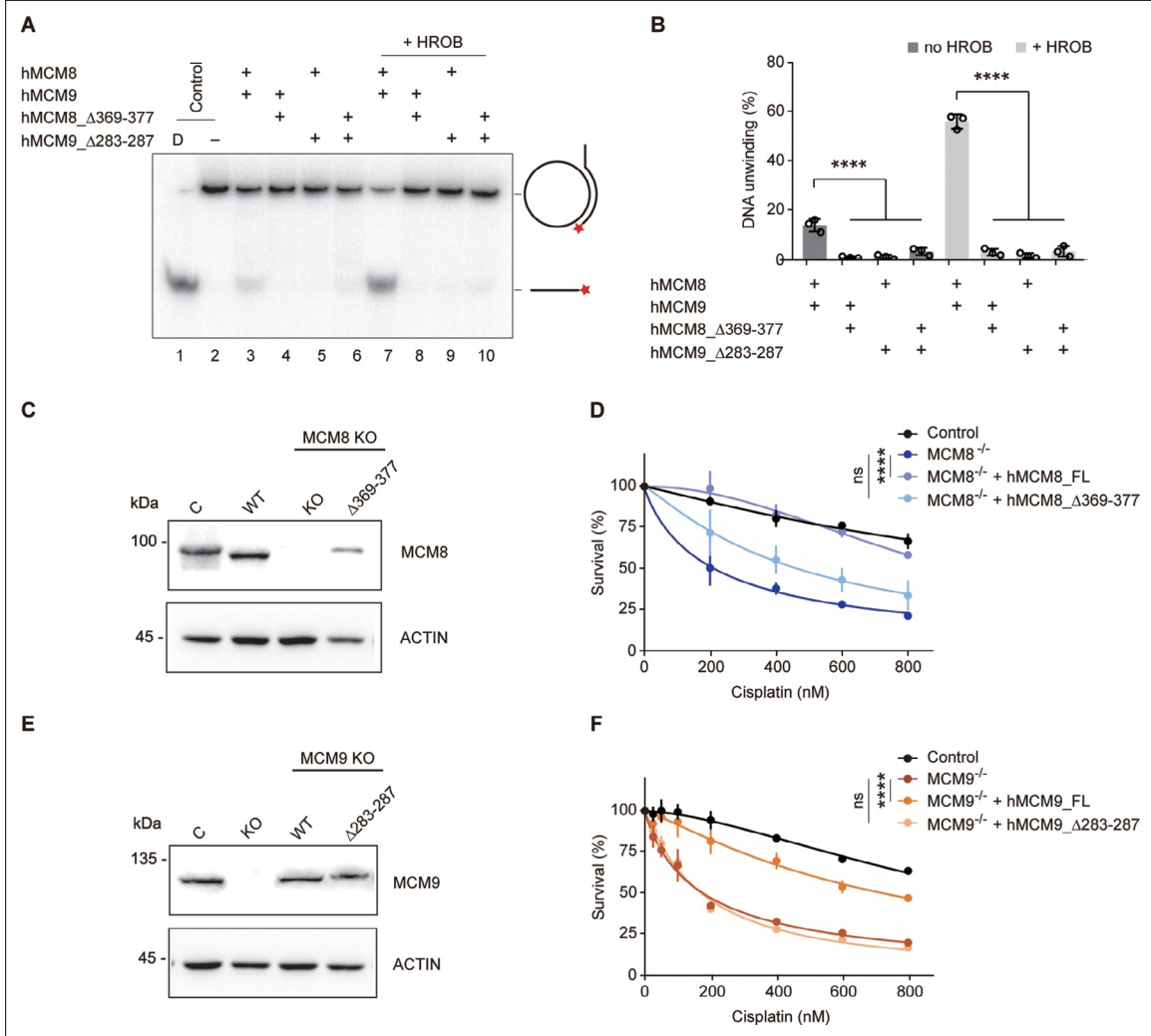

**Figure 6.** Analysis of the helicase activities and chemoresistance exhibited by the N-C linkers. (**A**) Representative autoradiograph of the DNA unwinding reaction was conducted using a $^{32}$P-labeled ssDNA oligo annealed to the M13mp18 as DNA substrate in the presence or absence of HROB with purified MCM8/9, MCM8 N-C linker mutant (MCM8Δ369-377) or MCM9 N-C linker mutant (MCM9Δ283-287), either alone or in combination. D-boiled DNA substrate control. (**B**) Graphical representation of the percentage of DNA unwinding in reactions conducted as in (**A**). The mean ± SD of three independent experiments is presented. Statistical analysis was conducted using one-way ANOVA (****p<0.0001). (**C**) Western blot to detect MCM8 in DT40 control cells or in an *MCM8* KO clone reconstituted with MCM8 WT or MCM8_Δ369–377 mutant. Actin is shown as a loading control. C, control cells; Δ369–377, MCM8_Δ369–377. (**D**) Survival analysis in DT40 control cell, MCM8 KO cell, or cells reconstituted with MCM8 WT or MCM8 N-C linker mutant upon treatment with cisplatin. Cell survival is expressed as a percentage of an untreated control. The mean ± SD of three independent experiments is presented. Statistical analysis was conducted on data points at four distinct cisplatin concentrations (200, 400, 600, 800 nM) using Student's t-test (****p<0.0001, at all four concentrations analyzed). (**E**) Detection by western blot of MCM9 in DT40 control cells or in an *MCM9* KO clone reconstituted with MCM9 WT or MCM9_Δ283–287 mutant. Actin is shown as a loading control. C, control cells; Δ283–287, MCM9_Δ283–287. (**F**) Survival analysis in DT40 control cell, MCM9 KO cell, or cells reconstituted with MCM9 WT, or MCM9 N-C linker mutant upon treatment with cisplatin. Cell survival is represented as in (**D**) and statistical analysis was conducted as in (**D**) (****p<0.0001, at all four concentrations analyzed).

The online version of this article includes the following source data and figure supplement(s) for figure 6:

**Source data 1.** The helicase activities of MCM8/9 N-C linker mutants.

**Source data 2.** Western blot to detect MCM8 or MCM9 in DT40 control cells or in an *MCM8/MCM9* KO clone.

**Figure supplement 1.** Subgroups of the N-C linkers of MCM2-9.

**Figure supplement 2.** SEC profiles of wild-type (WT) and N-C linker mutant of MCM8/9 complex.

on the OB-hps of MCM8 and MCM9 (F353 of hMCM8, F251 of hMCM9) (*Figure 5D*). They will most likely perform functions similar to those found in MCM2-7.

We also present a new conformation (Conformation II) of the MCM8/9 NTD ring induced by HROB, which may enhance the DNA binding ability of the MCM8/9 complex. We found that when MCM8/9 is complexed with the CTD domain of HROB, it becomes more stable and exhibits significantly higher helicase activity. The HROB CTD domain is about 20 kDa and is composed of a flexible loop and an OB-fold domain. The flexible loop containing ~20 amino acids is sufficient to interact with the MCM8/9 complex, but it alone is unable to stimulate the helicase activity of MCM8/9 (*Huang et al., 2020*). Our helicase assay demonstrated that the OB-fold domain of HROB is required for MCM8/9 activation. The OB-fold domain is a nucleic acid-binding motif that likely help to load the DNA into the central channel of MCM8/9. Induced by HROB, the structure of the MCM8/9 NTD ring convert its *C3* symmetry to *C1*, and two of the MCM8/9 heterodimers translate and tilt following the expansion of the trimer interface. When the MCM2-7 double hexamer encircles the double-stranded origin DNA, the MCM5-MCM2 interface characterizes as a 'gate' that undergoes open and closed conformation change to engage the DNA entry (*Samel et al., 2014*). The expansion of the MCM8/9 trimer interface may involve a transient conformation state to facilitate the DNA entry into the inner channel of the MCM8/9 hexamer. Thus, HROB not only aids in the loading of DNA into MCM8/9, but also acts as an activator, collaborating with MCM8/9 to promote DNA unwinding.

Benefiting from the rapid development of cryo-electron microscopy technology over the past decade, a dozen structures of MCM2-7-containing intermediates have been resolved including the OCCM, DH, and CMG with/ without DNA (*Yuan and Li, 2020*). Based on these structures, the hand-over-hand rotary mode and the 'dam-and-diversion tunnel' model had been proposed for MCM2-7 helicase in DNA translocation and unwinding (*Yuan and Li, 2020*; *Eickhoff et al., 2019*). For MCM8/9 helicase, there is currently no established model to describe its unwinding mechanism. Here, our structural and biochemical studies have provided some hints in this regard. First of all, the structural dynamics analysis reveals that the MCM8/9 CTD ring rotates relative to the NTD ring. Second, the length of the MCM8 and MCM9 N-C linkers are surprisingly much longer than those of the MCM2-7 complex, which may permit the rotations of MCM8/9. The fact that single-site structure-guided mutations on the ACL loops have no effect on the helicase activity of MCM8/9 further supports the idea that the N-C linkers, rather than the ACL loop, mediated the interaction between the N-tier ring and the C-tier ring of MCM8/9. Last but not least, HROB can convert inactive MCM8/9 into an active helicase by recruiting and inducing a conformational change in the MCM8/9 NTD ring.

Although a rotary model is in sight and our findings also hint at a distinct unwinding mechanism of MCM8/9, many questions remain unsolved. The most pressing issue is that the type of DNA substrate unwound by MCM8/9 is unknown. In vivo sequencing and in vitro DNA binding results are required to screen appropriate substrates. Furthermore, as the HROB only stimulates ~60% of the MCM8/9 helicase in vitro, we believe that other regulators remain to be discover for the full activation of MCM8/9 helicase. Based on these results, the high-resolution structures of MCM8/9 with DNA in the presence of ADP or ATP are necessary to fully clarify the unwinding mechanism of MCM8/9. Despite these limitations, the MCM8/9 complex structures provided in this study will be a rich source of information for further research into the mechanism of DNA helicase in homologous recombination, as well as for the analysis of multiple disease mutations in MCM8/9 associated with premature ovarian failure as well as cancers.

## Materials and methods
### Plasmids
The human MCM8/9 genes were obtained by reverse transcription of mRNA from HeLa cells. The gene fragments coding MCM8/9 of *Gallus gallus* were also from corresponding reverse transcripts and the total RNA was extracted from the chicken DT40 cell line. Human MCM8_61–840 and MCM9_1–684 were inserted into the pFastbac-Dual vector (Invitrogen). *Gallus gallus* MCM8_50–830 and MCM9_1–691 were inserted into the pFastbac-1 vector, respectively. All the MCM8 were fused with 6x His tag at their C-terminus for subsequent affinity purification. The genes coding hMCM8/9 NTDs (MCM8_61–376, MCM9_1–276) and relative mutants were cloned into multiple cloning site I (MCS I) of pRSFDuet-1 vector (Novagen). All of the mutations were introduced by the standard PCR-based

mutagenesis method described before (*Weng et al., 2019*). Human HROB (NM_001171251.3) was obtained by reverse transcription of mRNA from 293T cells. HROB-CTD_391–580 was inserted into the pET28S-SUMO vector which expresses the N-terminal SUMO tag. All constructs were confirmed by DNA sequencing.

## Protein expression and purification

To express the core hMCM8/9 and gMCM8/9 complex proteins, recombinant baculoviruses were prepared using the Bac-to-Bac expression system (Invitrogen). The proteins were expressed in *Trichoplusia ni* (BTI-Tn5B1–4, Hi5) insect cells for 60 hr at 27 °C before harvesting. The cell pellet was collected by centrifugation at 800 × g and then freshly frozen by liquid nitrogen and stored at –80 °C before use. For purification, the cell pellet was resuspended with lysis buffer (50 mM HEPES, pH 8.0, 350 mM KCl, 50 mM sodium glutamate, 10% glycerol, and 30 mM imidazole) and lysed by cell homogenizer (Avestin Emulsiflex C3). Then the lysate was clarified by centrifugation at 16,000 × rpm for 30 min. The supernatant was mixed with nickel-NTA resin (Novagen) and was continuously stirred for 1 hr at 4 °C for adequate binding. The beads were then collected into a column and extensively washed with lysis buffer to remove undesired proteins. Target proteins were then eluted down with elution buffer (50 mM HEPES, pH 7.8, 80 mM KCl, 20 mM sodium glutamate, 5% glycerol, and 300 mM imidazole). The proteins were further purified by HiTrap Heparin HP 5 ml column (GE Healthcare) and size exclusion column (Superose 6 Increase 10/300 GL, GE Healthcare) in buffer (20 mM HEPES, pH 7.8, 150 mM KCl, 1 mM DTT, and 5% glycerol). The strategies of purification of MCM8/9 mutants were the same as that used for the core hMCM8/9 complex.

For expression of HROB-CTD, 10 mL BL21 codon plus strain containing the recombinant pET28a-HROB-CTD plasmid was inoculated into 1 L LB medium. The bacteria solution was firstly cultured at 37 °C until the OD600 ≈ 0.8 and then cooled down to 16 °C. The expression of recombinant protein was induced by IPTG (isopropyl-β-d-thiogalactoside) at a concentration of 0.2 mmol/L and kept expressing for another 20 hr. Subsequently, the cells were harvested and disrupted by a high-pressure cell homogenizer in lysing buffer. After centrifugation, the supernatant was mixed with nickel-NTA beads and retained for 1 hr at 4 °C with rolling. Before elution, the SUMO tag was cleaved by Ulp Protease and finally, the HROB-CTD protein was eluted in the buffer containing 50 mM HEPES, pH 7.8, 80 mM KCl, 20 mM sodium glutamate, 5% glycerol. Following Ni-NTA affinity purification, the eluted protein was further purified by HiTrap Heparin HP 5 ml column (GE Healthcare) and size exclusion chromatography (HiLoadTM16/600 Superdex TM pg, GE Healthcare) in the same buffer using hMCM8/9. The GST tag HROB-CTD was also expressed in BL21 (DE3) *E. coli* cells and purified using GSH-Sepharose affinity chromatography (GE Healthcare) followed by the size-exclusion chromatography. And the HROB-MBM peptides (HROB_391–413) were commercially synthesized by China Peptide Co., Ltd.

## Cryo-EM sample preparation

The fresh gMCM8/9 proteins were used to prepare the cryo-EM sample. Briefly, 0.8 μL crosslinker BS(PEG)$_9$ (Thermo Scientific) and 10 μM protein were incubated for 1 hr at 4 °C and filtered by Superose 6 in buffer (20 mM HEPES, pH 7.8, 150 mM KCl) after centrifugation. The fractions were identified by SDS-PAGE and the peak fraction was diluted to about 0.3 mg/mL to prepare cryo-EM samples. 4 μL fresh sample was applied on glow-discharged Cu holey carbon grids (Quantifoil R 1.2/1.3) and incubated for 1 min at room temperature. Grids were then blotted for 3 s in 100% humidity at 4 °C and plunged frozen in liquid ethane cooled by liquid nitrogen using Vitrobot (Thermo).

For the hMCM8/9-HROB complex, the hMCM8/9 NTDs and HROB proteins were mixed by the ratio 1:3. 4 μL fresh sample (0.48 mg/mL) was applied on glow-discharged Au holey carbon grids (Quantifoil R 0.6/1) and the following operation was the same as described above for preparing the sample of gMCM8/9 complex.

## Cryo-EM data acquisition

The grids of gMCM8/9 were loaded onto a transmission electron microscope operated at 200 kV (FEI Talos Arctica) in liquid nitrogen temperatures. Images were recorded on a Gatan K2 camera in counting mode, at a nominal magnification of 29,000x. The total dose rate was about 50 e⁻/Å² for each micrograph stack. The total dose was fractionalized to 32 frames equally, corresponding to a physical

pixel size of 1.0 Å (0.5 Å super-resolution pixel size). The defocus range was set from −1.7 to −2.2 µm. 2891 raw movie micrographs were collected using the serialEM software (*Mastronarde, 2005*).

The grids of the hMCM8/9 NTD complex were loaded onto a transmission electron microscope operated at 300 kV (FEI Titan Krios) equipped with a K3 direct electron detector. The parameter settings of total dose and defocus range were as same as that of gMCM8/9. While the images' physical pixel size is 0.827 Å and the super-resolution pixel size is 0.4135 Å. 4078 raw movie micrographs were collected using the Thermo Fisher Scientific EPU software.

## Single particle data processing

The data set of gMCM8/9 was processed by Cryosparc-3.2 (*Punjani et al., 2017*). All the movies were processed using patch motion correction and patch CTF estimation. A total of 2,134,191 particles were automatically picked by the template picker and 1,574,821 particles were produced after the inspect particle picks treatment. Three parallel runs of 2D classification (K=100) were performed using the data (binned 4) with a box size of 256 pixels. A small subset of the 'good' particles with clear secondary-structure features was selected from 2D classification results and was used to generate four initial models. The 434,320 particles in the best initial model class were re-extracted with the original pixel size and further refined to a 6.7 Å low-resolution reconstruction using non-uniform refinement (New) (*Punjani et al., 2020*). Four resolution gradient templates generated from non-uniform refinement were used for the multi-reference heterogeneous refinement, resulting in a 4.31 Å global reconstruction (290,080 particles) renamed consensus map. The NTD map was improved to 3.73 Å with local refinement followed by applying a tight N-terminal mask with *C3* symmetry. Local CTF refinement further improved the resolution to 3.66 Å. For the CTD map, a reference was morphed from a predicted gMCM8/9 CTD model using AlphaFold (*Figure 1—figure supplement 7 Varadi et al., 2022*). Before processing CTD local refinement (applying *C3* symmetry), the projection of NTD particles from the 3.66 Å NTD map was subtracted from the 4.31 Å consensus map, and generated a 6.57 Å CTD hexamer map. Finally, 3D classification and local refinement resulted in a 5.21 Å (mean resolution, validation from Phenix *Liebschner et al., 2019*) CTD map using 72,948 particles. The 3.66 Å NTD map and 5.21 Å CTD map were aligned to the consensus map, respectively, and then merged together to generate a final map.

The EMDB accession code is EMD-32346 and the PDB entry ID is 7W7P. The image processing and 3D reconstruction steps were illustrated in *Figure 1—figure supplement 8*.

RELION-3.1.1 (*Scheres, 2012*) was also used to process the same data, resulting in a 4.2 Å NTD map and a 5.4 Å CTD map. All the movie frames were motion-corrected by MotionCorr2 (*Zheng et al., 2017*) with a patch alignment of 5 × 5. CTF parameters were estimated using Gctf (*Zhang, 2016*). 965,175 particles were auto-picked and extracted into 4x binned particles, which rescales the initial box size at 256 pixels to the final box size of 64 pixels. The particles were subjected to 2 rounds of 2D classifications (K=100). 388,378 particles with clear secondary-structure features were selected for 3D initial model generation. Three parallel runs of multi-reference 3D classification (K=9) were performed and the best class was merged. 144,202 particles were re-extracted at the original pixel size and a final map was imported as a reference and were submitted to the final 3D auto-refinement. Finally, we yielded a 6.6 Å global 3D density map estimated by the gold-standard Fourier shell correlation at a correlation cutoff value of 0.143.

As the density of the gMCM8/9 CTD was a little blurred, we then used the multi-body refinement program (*Nakane et al., 2018*) to process the dataset. 6.6 Å gMCM8/9 global map's NTD and CTD were subtracted as the first and the second subtracted body, respectively, using Chimera (*Pettersen et al., 2004*). The corresponding masks were generated from the final map using mask creation in RELION. The multi-body refinement resulted in a 4.2 Å 3D density map of gMCM8/9 NTD and a 5.4 Å 3D density map of gMCM8/9 CTD. It generated 12 components, with each including a discrete number of independently moving bodies. The first five components explained 86.77% of the variance in the data. The image processing and 3D reconstruction steps were illustrated in *Figure 1—figure supplement 9*.

The data set of the MCM8/9 NTD complex was processed by Cryosparc-3.3. In brief, a total of 1,855,585 particles were automatically picked by the template picker, and 677,960 particles were produced after the inspect particle picks treatment. After the processing of 2D classification, Ab-Initio Reconstruction, and Heterogeneous refinement (K=3), the 95,257 particles in the best 3D class

were re-extracted with original pixel size and further refined to a 3.95 Å resolution reconstruction (C1 symmetry) using Non-uniform refinement (New). The image processing and 3D reconstruction steps were illustrated in *Figure 4—figure supplement 2*.

## Model building and refinement

For the gMCM8/9 model building, the gMCM8 NTD and gMCM9 NTD monomer models were predicted through SWISS-MODEL by providing the hMCM8 NTD (PDB: 7DP3) and hMCM9 NTD (PDB: 7DPD) structural template, respectively. First, the predicted structures were used as the initial model directly docking into the EM map. As the gMCM8/9 NTD has a *C3* symmetry axis, another two copies were generated by horizontal rotating for subsequent rigid body-fitting into the 3D density map with the program Chimera. Second, using the automatic refinement plugin function in COOT (*Emsley et al., 2010*) to refine the initial model. The final model was refined against the corresponding map using PHENIX (*Afonine et al., 2012*) in real space with secondary structure and geometry restraints. The gMCM8/9 CTD hexamer model was predicted as above described and the processing of docking and refinement was the same as the gMCM8/9 NTD model.

For the hMCM8/9 NTD complex model building, in brief, hMCM8 NTD (PDB: 7DP3) and hMCM9 NTD (PDB: 7DPD) models were used as the initial model directly docking into the EM map. The other processing procedure was similar to that of the gMCM8/9 model building. All the structures of the gMCM8/9 complex were validated through examination of the Molprobity scores and statistics of the Ramachandran plots. Molprobity scores were calculated as described in *Table 2*.

## Helicase assays

To prepare the substrate, the oligonucleotide (5'-(dT)$_{40}$GTTTTCCCAGTCACGACG-TTGTAAAACGAC GGCCAGTGCC-3') containing a 40 nt region complementary to the M13mp18(+) strand and a 40 nt oligo-dT at the 5' end was labeled at the 3' terminus with [α-$^{32}$P] dCTP (Perkin Elmer) and annealed to the single-stranded DNA M13mp18 (*Huang et al., 2020*). 0.1 nM (in molecules) DNA substrates were mixed with 5 µg recombinant MCM8/9 complex as well as its mutants as indicated within each 15 µl volume reaction in the helicase buffer (25 mM HEPES, pH 7.5, 1 mM magnesium acetate, 25 mM sodium acetate, pH 5.2, 4 mM ATP, 0.1 mg/ml BSA, 1 mM DTT). 2.5 µg HROB was used as an activator. To avoid re-annealing, the reaction was supplemented with a 100-fold unlabeled oligonucleotide. The reactions were then incubated at 37 °C for 60 min and stopped by adding 1 µl of stop buffer (0.4% SDS, 30 mM EDTA, and 6% glycerol) and 1 µl of proteinase K (20 mg/ml, Sigma) into the reaction for another 10 min incubation at 37 °C. The products were separated by 15% polyacrylamide gel electro-phoresis in 1x TBE buffer and analyzed by the Amersham typhoon (Cytiva).

## Cell-viability assay

Cell-viability assays were performed as described previously (*Nishimura et al., 2012*). Briefly, the *MCM8* or *MCM9* KO DT40 cell lines were transfected with plasmids containing wild-type MCM8, MCM9, or the indicated mutants by electroporation. $1.0 \times 10^3$ cells were seeded into each well of a 96-well plate and treated with a range of concentrations of cisplatin. After 48 hr of incubation, the cell viability was measured by the Cell counting kit 8 (CCK8) according to the standard procedures. The absorbance readings at 405 nm were conducted using an Epoch Microplate Spectrophotometer (BioTek Instrumentals, Inc, Winooski, VT, USA). All experiments were performed at least three times.

## Pull down assay

The HEK293T cells transfected with Flag-hMCM8/9-FL or Flag-hMCM8/9-NTD were cultured over-night and washed twice with cold phosphate-buffered saline (PBS). Cell pellets were resuspended with lysis buffer (20 mM Tris, pH7.5, 150 mM NaCl, 5 mM EDTA, 0.5% NP-40, 10% glycerol, protease inhibitor cocktail (Roche, 04693132001)). After incubation for 45 min at 4 °C with gentle agitation, the whole-cell lysates were collected by centrifugation (12,000 × g for 15 min, at 4 °C). GST beads coupled with 2 µg GST-HROB or GST alone were then incubated with an equal volume of above HEK293T cell lysates at 4 °C for 4 hr. The beads were washed four times with lysis buffer. Proteins bound to the beads were separated by SDS–PAGE and subsequently immunoblotted with anti-Flag antibody (Affinity Biosciences, Cat #125243).

## Cell lines

Cell lines of HEK293T (ATCC, CRL-3216) and DT40 (ATCC, CRL-2111) were purchased from the American Type Culture Collection (ATCC) and maintained in Dulbecco's modified Eagle's medium (DMEM, GIBCO, C11965500CP) supplemented with 10% fetal bovine serum (FBS, GIBCO, 10270106). All the cell lines have been tested negative for mycoplasma contamination before experiments.

## Statistical analysis

The statistical analysis was performed using Prism (GraphPad Software Inc) on the data from at least three independent experiments. Statistical significance was assessed by a two-tailed unpaired Student's $t$-test. $p \leq 0.05$ was considered significant.

## Acknowledgements

We thank the Cryo-EM Centre, Southern University of Science and Technology for assistance in cryo-EM data collection. We are grateful to Hongjie Zhang (Institute of Biophysics, CAS) for technical assistance in helicase assays. Funding The work was supported by the grants from Shenzhen Science and technology planning project (project No. JCYJ20200109142412265, ZDSYS20220606100803007 to YL, project No.RCBS20200714114922284 to ZW); 'Pearl River Talents Plan' Innovation and Entrepreneurship Team Project of Guangdong Province (project No.2019ZT08Y464 to YL), National Key R&D Program of China (project No. 2022YFE0210000 to YL) ; National Natural Science Foundation of China (project No.32000860, 32271321 to ZW) and Natural Science Foundation of Guangdong Province (project No.2023A1515010245 to ZW)

## Additional information

### Funding

| Funder | Grant reference number | Author |
| --- | --- | --- |
| Shenzhen Scientific and Technological Foundation | JCYJ20200109142412265 | Yingfang Liu |
| Shenzhen Key Laboratory Fund | ZDSYS20220606100803007 | Yingfang Liu |
| Shenzhen Scientific and Technological Foundation | RCBS20200714114922284 | Zhuangfeng Weng |
| Pearl River Talent Plan of Guangdong | 2019ZT08Y464 | Yingfang Liu |
| National Natural Science Foundation of China | 32000860 | Zhuangfeng Weng |
| National Key Research and Development Program of China | 2022YFE0210000 | Yingfang Liu |
| National Natural Science Foundation of China | 32271321 | Zhuangfeng Weng |
| Natural Science Foundation of Guangdong Province | 2023A1515010245 | Zhuangfeng Weng |

The funders had no role in study design, data collection and interpretation, or the decision to submit the work for publication.

### Author contributions

Zhuangfeng Weng, Conceptualization, Funding acquisition, Investigation, Visualization, Writing - original draft, Writing - review and editing; Jiefu Zheng, Software, Investigation, Visualization, Writing - original draft; Yiyi Zhou, Investigation; Zuer Lu, Investigation, Visualization; Yixi Wu, Data curation, Software; Dongyi Xu, Supervision; Huanhuan Li, Validation, Visualization, Methodology, Writing

- original draft; Huanhuan Liang, Conceptualization, Supervision, Project administration, Writing - review and editing; Yingfang Liu, Supervision, Funding acquisition, Project administration, Writing - review and editing

## Author ORCIDs
Jiefu Zheng (iD) http://orcid.org/0000-0002-8716-810X
Dongyi Xu (iD) http://orcid.org/0000-0001-5711-2618
Yingfang Liu (iD) http://orcid.org/0000-0002-1835-0397

Reviewer #1 (Public Review): https://doi.org/10.7554/eLife.87468.3.sa1
Reviewer #2 (Public Review): https://doi.org/10.7554/eLife.87468.3.sa2
Author Response https://doi.org/10.7554/eLife.87468.3.sa3

---

# Additional files

## Supplementary files
• MDAR checklist

## Data availability
The 3D cryo-EM maps have been deposited to the EMDB database with accession numbers: EMD-32346 and EMD-33989. The atomic models have been deposited at the RSCB PDB under the accession codes 7W7P and 7YOX. All data generated or analyzed during this study are included in the manuscript and supporting files. Source Data files have been provided for Figure 4, 5 and 6.

The following datasets were generated:

The following dataset was generated:

| Author(s) | Year | Dataset title | Dataset URL | Database and Identifier |
|---|---|---|---|---|
| Zheng JF, Weng ZF, Liu YF | 2023 | Cryo-EM structure of gMCM8/9 helicase | https://www.ebi.ac.uk/emdb/EMD-32346 | Electron Microscopy Data Bank, EMD-32346 |
| Liu YF, Weng ZF, Zheng JF | 2023 | Cryo-EM structure of the N-terminal domain of hMCM8/9 and HROB | https://www.ebi.ac.uk/emdb/EMD-33989 | Electron Microscopy Data Bank, EMD-33989 |
| Zheng JF, Weng ZF, Liu YF | 2023 | Cryo-EM structure of gMCM8/9 helicase | https://www.rcsb.org/structure/7W7P | RCSB Protein Data Bank, 7W7P |
| Zheng JF, Weng ZF, Liu YF | 2023 | Cryo-EM structure of the N-terminal domain of hMCM8/9 and HROB | https://www.rcsb.org/structure/7YOX | RCSB Protein Data Bank, 7YOX |

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
