## [Editor Report · eLife assessment]

This paper presents **important** findings on the hexametric structure of MCM8/9, which potentially explain its role as a DNA helicase in homologous recombination. This **solid** work will be of interest to biologists studying DNA transactions.

---

## [Referee Report · Reviewer #1 (Public Review)]

MCM8 and MCM9 are paralogues of the eukaryotic MCM2-7 proteins. MCM2-7 form a heterohexameric complex to function as a replicative helicase while MCM8-9 form another hexameric helicase complex that may function in homologous recombination-mediated long-tract gene conversion and/or break-induced replication. MCM2-7 complex is loaded during the low Cdk period by ORC, CDC6, and Cdt1, when the origin DNA may intrude into the central channel via the MCM2-MCM5 entry "gate". In the S phase, MCM2-7 complex is activated as CMG helicase with the help of CDC45 and GINS complex. On the other hand, it still remains unclear how MCM8-9 complex is loaded onto DNA and then activated.

In this study, the authors first investigated the cryo-EM structure of chicken MCM8-9 (gMCM8-9) complex. Based on the data obtained, they suggest that the observed gMCM8-9 structure might represent the structure of a loading state with possible DNA entry "gate". The authors further investigated the cryo-EM structure of human MCM8-9 (hMCM8-9) complex in the presence of the activator protein, HROB, and compared the structure with that obtained without HROB1, which the authors published previously. As a result, they suggest that MCM8-9 complex may change the conformation upon HROB binding, leading to helicase activation. Furthermore, based on the structural analyses, they identified some important residues and motifs in MCM8-9 complex, mutations of which actually impaired the MCM8-9 activity in vitro and in vivo.

Overall, the data presented would support the authors' conclusions and would be of wide interest for those working in the fields of DNA replication and repair. One caveat is that most of the structural data are shown only as ribbon model without showing the density map data obtained by cryo-EM, which makes accurate evaluation of the data somewhat difficult.

Addition after review of the revised manuscript: The authors have made a reasonable attempt to address the points raised by the reviewers, by which the paper is significantly improved.

---

## [Referee Report · Reviewer #2 (Public Review)]

MCM8 and MCM9 together form a hexameric DNA helicase that is involved in homologous recombination (HR) for repairing DNA double-strand breaks. The authors have previously reported on the winged-helix structure of the MCM8 (Zeng et al. BBRC, 2020) and the N-terminal structure of MCM8/9 hexametric complex (MCM8/9-NTD) (Li et al. Structure, 2021). This manuscript reports the structure of a near-complete MCM8/9 complex and the conformational change of MCM8/9-NTD in the presence of its binding protein, HROB, as well as the residues important for its helicase activity.

The presented data might potentially explain how MCM8/9 works as a helicase. However, additional studies are required to conclude this point because the presented MCM8/9 structure is not a DNA-bound form and HROB is not visible in the presented structural data. Taking into these accounts, this work will be of interest to biologists studying DNA transactions.

A strength of this paper is that the authors revealed the near-complete MCM8/9 structure with 3.66A and 5.21A for the NTD and CTD, respectively (Figure 1). Additionally, the authors discovered a conformational change in the MCM8/9-NTD when HROB was included (Figure 4) and a flexible nature of MCM8/9-CTD (Figure S6 and Movie 1).

The revised version of "Structural and mechanistic insights into the MCM8/9 helicase complex" by Weng et al. includes only very minor changes in the text and incorporates two additional supplementary figures (S8 and S11) illustrating the size of MCM8/9 mutants.

In the previous version, I raised two important concerns that required addressing. (1) The presented structures exclusively depicted the unbound forms of DNA. It is crucial to elucidate the structure of a DNA-bound form. (2) The MCM8/9 activator, HROB, was not visible in the structural data. Although HROB induced a conformational change in MCM8/9-NTD, it is essential to visualize the structure of an MCM8/9-HROB complex.

The authors neither addressed nor provided new data in response to these issues. Consequently, I maintain my initial stance and have no further comments on the revised version.

---

## [Author Response]

The following is the authors’ response to the original reviews.

**Reviewer #1 (Public Review):**
MCM8 and MCM9 are paralogues of the eukaryotic MCM2-7 proteins. MCM2-7 form a heterohexameric complex to function as a replicative helicase while MCM8-9 form another hexameric helicase complex that may function in homologous recombination-mediated longtract gene conversion and/or break-induced replication. MCM2-7 complex is loaded during the low Cdk period by ORC, CDC6, and Cdt1, when the origin DNA may intrude into the central channel via the MCM2-MCM5 entry "gate". In the S phase, MCM2-7 complex is activated as CMG helicase with the help of CDC45 and GINS complex. On the other hand, it still remains unclear how MCM8-9 complex is loaded onto DNA and then activated.In this study, the authors first investigated the cryo-EM structure of chicken MCM8-9 (gMCM89) complex. Based on the data obtained, they suggest that the observed gMCM8-9 structure might represent the structure of a loading state with possible DNA entry "gate". The authors further investigated the cryo-EM structure of human MCM8-9 (hMCM8-9) complex in the presence of the activator protein, HROB, and compared the structure with that obtained without HROB1, which the authors published previously. As a result, they suggest that MCM8-9 complex may change the conformation upon HROB binding, leading to helicase activation. Furthermore, based on the structural analyses, they identified some important residues and motifs in MCM8-9 complex, mutations of which actually impaired the MCM8-9 activity in vitro and in vivo.Overall, the data presented would support the authors' conclusions and would be of wide interest for those working in the fields of DNA replication and repair. One caveat is that most of the structural data are shown only as ribbon model without showing the density map data obtained by cryo-EM, which makes accurate evaluation of the data somewhat difficult.

We thank the reviewer for the positive comments on our work. For evaluating all the structural data, in our revised manuscript, we have presented the density maps of the cryo-EM structures of the gMCM8/9 complex in supplementary figure S5 and S6. In addition, the 3D cryo-EM map of the gMCM8/9 complex and the hMCM8/9 NTD ring have been deposited to the EMDB database with accession number EMD-32346 and EMD-33989, respectively. The corresponding atomic models have been deposited at the RSCB PDB under the accession code 7W7P and 7YOX, respectively. All these data have been released in May 2023.

**Reviewer #2 (Public Review):**
MCM8 and MCM9 together form a hexameric DNA helicase that is involved in homologous recombination (HR) for repairing DNA double-strand breaks. The authors have previously reported on the winged-helix structure of the MCM8 (Zeng et al. BBRC, 2020) and the Nterminal structure of MCM8/9 hexametric complex (MCM8/9-NTD) (Li et al. Structure, 2021). This manuscript reports the structure of a near-complete MCM8/9 complex and the conformational change of MCM8/9-NTD in the presence of its binding protein, HROB, as well as the residues important for its helicase activity.The presented data might potentially explain how MCM8/9 works as a helicase. However, additional studies are required to conclude this point because the presented MCM8/9 structure is not a DNA-bound form and HROB is not visible in the presented structural data. Taking into these accounts, this work will be of interest to biologists studying DNA transactions.A strength of this paper is that the authors revealed the near-complete MCM8/9 structure with 3.66A and 5.21A for the NTD and CTD, respectively (Figure 1). Additionally, the authors discovered a conformational change in the MCM8/9-NTD when HROB was included (Figure 4) and a flexible nature of MCM8/9-CTD (Figure S6 and Movie 1).The biochemical data that demonstrate the significance of the Ob-hp motif and the N-C linker for DNA helicase activity require careful interpretation (Figures 5 and 6). To support the conclusion, the authors should show that the mutant proteins form the hexamer without problems. Otherwise, it is conceivable that the mutant proteins are flawed in complex formation. If that is the case, the authors cannot conclude that these motifs are vital for the helicase function.A weakness of this paper is that the authors have already reported the structure of MCM8/9NTD utilizing human proteins (Li et al. Structure, 2021). Although they succeeded in revealing the high-resolution structure of MCM8/9-NTD with the chicken proteins in this study, the two structures are extremely comparable (Figure S2), and the interaction surfaces seem to be the same (Figure 2).Another weakness of this paper is that the presented data cannot fully elucidate the mechanistic insights into how MCM8/9 functions as a helicase for two reasons. (1) The presented structures solely depict DNA unbound forms. It is critical to reveal the structure of a DNA-bound form. (2) The MCM8/9 activator, HROB, is not visible in the structural data. Even though HROB caused a conformational change in MCM8/9-NTD, it is critical to visualize the structure of an MCM8/9HROB complex.

We appreciate the reviewer’s comments on our work. Regarding the first weakness mentioned above, the previously reported cryo-EM structure of hMCM8/9 NTD ring was achieved with a resolution of 6.6 Å. At this level of resolution, we were only able to observe the overall shape of the structure and a partial representation of the protein's secondary structure. It is hard for us to discern any specific details regarding the interaction interface between MCM8 and MCM9. In this study, we solved the structure of gMCM8/9 NTD ring with a resolution of 3.67 Å. We believe that the higher resolution of gMCM8/9 NTD structure provides a significant advantage in analyzing the interaction surface between MCM8 and MCM9. This improved resolution has enabled us to gain valuable insights into the assembly mechanism of the MCM8/9 hexamer, representing a significant step forward in our understanding of the MCM8/9 helicase complex.In response to the second weakness raised by the reviewer, we fully agree with the reviewer that high-resolution structures of the MCM8/9 complex with DNA or HROB are necessary to elucidate the mechanism of this helicase complex. We are actively working towards obtaining these complex structures using cryo-EM and X-ray crystal diffraction.

Moreover, we would like to address the reviewer's concern regarding the mutant proteins used in the in vitro helicase assays. We have conducted additional experiments to confirm that these mutant proteins do not impair the formation of the MCM8/9 hexamer. Specifically, we performed size exclusion chromatography (SEC) analyses of the wild-type (WT) MCM8/9 complex, as well as MCM8 and MCM9 mutant proteins (Author response image 1). The results demonstrated that all the proteins behaved consistently and displayed similar SEC profiles during the purification process. Notably, the N-C linker deletion mutant (hMCM8_Δ369-377+MCM9_Δ283-287) combining the MCM8 and MCM9 N-C linker deletions also behaved similarly with WT MCM8/9 (Author response image 2). These findings strongly suggest that the mutations in the OB-hps regions and the N-C linkers do not disrupt the hexamer formation of the MCM8/9 complex. Author response image 1 and Author response image 2 have been included into the supplementary figure S8 and S11, respectively.

**Author response image 1. sa3fig1:** SEC profiles of WT and OB-hps mutants of MCM8/9 complex.

**Author response image 2. sa3fig2:** SEC profiles of WT and N-C linker mutant of MCM8/9 complex.

**Reviewer #1 (Recommendations For The Authors):**
I would like to provide some suggestions to improve the manuscript.1. Throughout the manuscript, more density map data obtained by the cryo-EM should be shown for accurate evaluation of the data. For example, in Figure 1C, the authors state that inner channel of the gMCM8-9 hexamer is ~28 angstrom, apparently based on the ribbon model. This is not appropriate because the space upon ribbon model is not same as that upon the density map. For Figure 1B, they state that "The domain structures of gMCM8-9 fit well into their electron map". If so, please show the actual docking data. Also for Figure 2, the docking presentation between the side chains in the ribbon model and the density map should be shown.

We sincerely appreciate the reviewer for the constructive suggestions. In addition to releasing our structural data in the EMDB and PDB, we have also followed the reviewer’s suggestions to included more density map data in the supplementary material. In fact, when calculating the dimeter of the inner channel of the MCM8/9 hexamer, we also measured that upon the density map (Author response image 3. A and B), which is consistent with our report in our manuscript. To further evaluate the structure of MCM8/9, we have included additional docking structures based on the density map (Author response image 3. C-F). Moreover, for Figure 2, more docking presentation are provided and the key residues involved in the hydrophobic interactions were highlighted in a bold manner (Author response image 4).Author response image 3 and Author response image 4 have been included into the supplementary figure S5 and S6, respectively.

**Author response image 3. sa3fig3:** The cryo-EM structure of gMCM8/9. (**A** and **B**) Reconstructed cryo-EM map of gMCM8/9.The diameter of the inner channel of MCM8/9 was measured at ~28 Å. (**C-F**) Representative regions of the cryo-EM structure of gMCM8/9 NTD are shown based on their density map. (**C**), chain A (MCM9); (**D**), chain B (MCM8); (**E**), chain C (MCM9); (**F**), chain D (MCM8).

**Author response image 4. sa3fig4:** Representative regions of the cryo-EM structure of gMCM8/9 NTD. (**A** and **B**), the region mediated hydrophobic interaction in figure 2B. (**A**) (MCM8), (**B**) (MCM9). (**C** and **D**), the region mediated hydrophobic interaction in figure 2C. (**C**) (MCM8), (**D**) (MCM9). The key residues were in bold.

1. Figures 4, 5, and 6: For helicase assay, more detailed experimental conditions (e.g. concentrations of DNA substrates and proteins used) should be presented. In addition, it should be described how Flag-hMCM8-9 complex (Figure 4C) was purified.

We sincerely appreciate the constructive suggestion provided by the reviewer. In the revised manuscript, we have included more experimental details in the helicase assays, including the concentrations of DNA substrates and proteins. The following paragraph describes the updated experimental procedure and also provided in the revise version of the manuscript.

Helicase assays: To prepare the substrate, the oligonucleotide (5'(dT)40GTTTTCCCAGTCACGACG-TTGTAAAACGACGGCCAGTGCC-3') containing a 40 ntregion complementary to the M13mp18(+) stand and a 40 nt oligo-dT at the 5′ end was labeled at the 3′ terminus with [α-32P] dCTP (Perkin Elmer) and annealed to the single-stranded DNA M13mp18 (24). 0.1 nM (in molecules) DNA substrates were respectively mixed with 5 µg recombinant MCM8/9 complex and its mutants as indicated within each 15 µl volume reaction in the helicase buffer (25 mM HEPES, pH 7.5, 1 mM magnesium acetate, 25 mM sodium acetate, pH 5.2, 4 mM ATP, 0.1 mg/ml BSA, 1 mM DTT). 2.5 µg HROB was used as an activator. To avoid re-annealing, the reaction was supplemented with a 100-fold unlabeled oligonucleotide. The reactions were then incubated at 37 °C for 60 min and stopped by adding 1 µl of stop buffer (0.4% SDS, 30 mM EDTA, and 6% glycerol) and 1µl of proteinase K (20 mg/ml, Sigma) into the reaction for another 10 min incubation at 37 °C. The products were separated by 15% polyacrylamide gel electrophoresis in 1× TBE buffer and analyzed by the Amersham typhoon (Cytiva).

In addition, to describe the expression of Flag-hMCM8/9 complex in Figure 4C, we have included the Pull-Down Assay in the “Material and Methods” section. The description is as follow: The HEK293T cells transfected with Flag-hMCM8/9-FL or Flag-hMCM8/9-NTD were cultured overnight and washed twice with cold phosphate-buffered saline (PBS). Cell pellets were resuspended with lysis buffer (20 mM Tris, pH7.5, 150 mM NaCl, 5mM EDTA, 0.5% NP-40, 10% glycerol, protease inhibitor cocktail (Roche, 04693132001)). After incubation for 45 min at 4°C with gentle agitation, the whole-cell lysates were collected by centrifugation (12,000 × g for 15 min, at 4 °C). GST beads coupled with 2 μg GST-HROB or GST alone were then incubated with an equal volume of above HEK293T cell lysates at 4°C for 4h. The beads were washed four times with lysis buffer. Proteins bound to the beads were separated by SDS–PAGE and subsequently immunoblotted with anti-Flag antibody (Cytiva).

1. Figure 3C: This is just an assumed model. Please clearly state it in the manuscript.

We appreciate the reviewer’s comment. We guess the reviewer is referring to Figure 5C. As Figure 3C depicts the top view of the gMCM8/9 hexamer structurally aligned with the MCM2-7 double hexamer (wheat) by aligning their respective C-tier ring. On the other hand, Figure 5C represents an assumed model where we docked a forked DNA fragment into the central channel of the gMCM8/9 hexamer. To address this assumed model, we have made thefollowing clarification in the revised manuscript: “We artificially docked a forked DNA into the central channel to generate a gMCM8/9-DNA model and found that the OB-hps of gMCM8 are capable to closely contact with it and insert their highly positively charged terminal loops into the major or minor grooves of the DNA strand, implying that they could be involved in substrateDNA processing and/or unwinding (Figure 5C)”.

1. Figure S1, C and D: The coloring of the gMCM8-9 CTD appears to show higher resolution than the NTD. May this be mispresentation?

We appreciate the reviewer's valuable feedback, and we have thoroughly re-evaluated Figure S1C and D. At the beginning, the local resolution distributions of the gMCM8/9 NTD and gMCM8/9 CTD were calculated using CryoSPARC. Upon re-examination, we found that the density maps of the gMCM8/9 CTD may be lower than 3.66 Å, because the density map of the gMCM8/9 CTD does not reveal more structural details than what is observed in the gMCM8/9 NTD. Thus, although the map shown in Figure S1D may appear to show a greater distribution of high-resolution regions., we would like to clarify that this discrepancy could be attributed to an optical illusion. We thank the reviewer for bringing this to our attention.

1. Figure S9: Is the "mean resolution" 5.21 angstrom identical to the Gold standard FSC? If not, please estimate the resolution using FSC, like other maps in this paper.

We thank the reviewer for the constructive suggestion. In response to this feedback, we would like to clarify the resolution estimation process for the gMCM8/9 CTD. Initially, we calculated the resolution of the gMCM8/9 CTD using the gold standard Fourier shell correlation (FSC) method, which yielded a resolution of 3.66 Å. However, upon further analysis, we identified an issue with the GSFSC Resolution curves, which led to an overestimation of the resolution based on the density map of the gMCM8/9 CTD. To ensure a more reliable and accurate estimation, we employed the Phenix software package to calculate the mean resolution during the refinement process of the gMCM8/9 CTD structure. The calculated mean resolution was determined to be 5.21 Å, which aligns more reasonably with the characteristics of the density map. To address any potential misunderstandings and provide clarity, we have explicitly labeled and described the evaluation process for this mean resolution in the "Single particle data processing" section of the Materials and Methods.

Minor points:1. Throughout the manuscript, there are several typographical and grammatical errors, which should be corrected. For example, in "Introduction", "GNIS complex" should be "GINS complex".

We thank the reviewer for pointing out the typographical and grammatical errors. We have corrected the grammar errors and polished our manuscript with the help of native speakers.

**Reviewer #2 (Recommendations For The Authors):**
1. "During HR repair, MCM8/9 was rapidly recruited to the DNA damage sites and colocalized with the recombinase Rad51 (21). It also interacted with the nuclease complex MRN (MRE11RAD50-NBS1) and was required for DNA resection at DSBs to facilitate the HR repair (Introduction)."There is a debate about whether MCM8/9-HROB colocalizes with RAD51 and whether it works upstream or downstream of RAD51 (Park et al. MCB, 2013; Lee et al. Nat Commun., 2015;Lutzmann et al. Mol Cell, 2012; Nishimura et al. Mol Cell, 2012; Natsume et al. G&D, 2017; Hustedt et al. G&D, 2019; Huang et al. Nat Commun., 2020).

We completely agree with the reviewer that previous studies have reported contradictory results regarding to the function of MCM8/9 in homologous recombination. Based on the structure information of MCM8/9, now we do not have direct evidence to resolve the ongoing debate. Nonetheless, based on our findings, we speculate that the MCM8/9 complex is likely involved in multiple steps within the process of homologous recombination. The structural insights provided by our study serve as a foundation for further investigations and may contribute to a better understanding of the complex and multifaceted roles of MCM8/9 in homologous recombination repair.

1. I noted that the BioRxiv version 1 (https://www.biorxiv.org/content/10.1101/2022.01.26.477944v1?versioned=true) contains a near-complete MCM8/9 with human protein based on the crystal analysis. Because its structure is comparable to chicken MCM8/9 revealed by cryo-EM, I highly suggest including this data in the manuscript.

We would like to thank the reviewer for this suggestion. The resolution of the hMCM8/9 crystal structure presented in our previous BioRxiv version is 6.6 Å, which is a little low. Moreover, it cannot provide more information than the present cryo-EM structures of MCM8/9. We are dedicated to optimizing the crystal quality and implementing strategies to enhance the resolution of the structure. We hope to present an improved crystal structure of hMCM8/9 in our forthcoming article.